# Direct Passive Participation: Aiming for Accuracy and Citizen Safety in the Era of Big Data and the Smart City

Ken Dooley [1,2]

1 The Department of Electrical Engineering and Automation, Aalto University, 00076 Espoo, Finland; kenneth.dooley@aalto.fi
2 Empathic Building, Haltian, 00520 Helsinki, Finland

**Abstract:** The public services in our smart cities should enable our citizens to live sustainable, safe and healthy lifestyles and they should be designed inclusively. This article examines emerging data-driven methods of citizen engagement that promise to deliver effortless engagement and discusses their suitability for the task at hand. Passive participation views citizens as sensors and data mining is used to elicit meaning from the vast amounts of data generated in a city. Direct passive participation has a clear link between the creation and the use of the data whereas indirect passive participation does not require a link between creation and use. The Helsinki city bike share scheme has been selected as a case study to further explore the concept of direct passive participation. The case study shows that passive user generated data is a strong indicator of optimum city bike station sizing relative to the existing methods that are already in use. Indirect passive participation is an important area of development; however, it still needs to be developed further. In the meantime, direct passive participation can be one of the tools used to design inclusive services in a way that is safe and an accurate representation of the citizens' needs.

**Keywords:** passive participation; citizen engagement; data-driven design; digital citizenship; smart city data

## 1. Introduction

According to the United Nations Sustainable Development Goals (SDGs) the best cities are inclusive, safe, resilient and sustainable. In order to achieve inclusiveness, city governments need to engage with their citizens and the methods of engagement are rapidly evolving in the digital age. Engagement is essential in order to provide public services that match the unique needs of the citizens in each neighborhood, district or city that is under analysis. The public service that is being developed cannot just be copied and pasted based on a what has worked in previous cases. One of the emerging methods is the use of the digital footprint data that is generated as citizens move through the city in their daily lives. Data from many different sources are being used more and more for engagement and in this article, we examine if its collection is safe for citizens and if it is an accurate representation of the needs. The SDGs make up the core of the 2030 Agenda for Sustainable Development which was adopted by all United Nations member states in 2015. The SDGs are a list of actions that target peace and prosperity for people and the planet. With this aim in mind, Smart Cities and Communities is listed as Sustainable Development Goal 11: Make cities and human settlements inclusive, safe, resilient and sustainable. At the heart of SDG 11, is the aim to improve life in cities by inclusively providing housing, transportation and other basic services. As part of this, the goal requires the service provision to be designed via direct, regular and democratic participation with citizens. SDG 11 also aims to reduce environmental impact and improve health and wellbeing in cities.

Arnstein's [1] seminal article introduced a 'ladder of participation' which defined steps to better engagement. The bottom steps are referred to as non-participation and represent a

predominantly one-way form of communication from decision makers to citizens. The next steps are referred to as degrees of tokenism and represent citizen consultation processes that are relatively passive and only ask for people's opinions without necessarily engaging them in debate. The top steps are called degrees of citizen power and the top step itself is referred to as citizen control. Participation is typically used in relation to a process which allows people to participate in decision making by offering their ideas and views whereas engagement goes further. Engagement suggests an interactive two-way process of discussion and dialogue that ensures that people's views inform a decision, alongside those of the expert and/or decision-maker. However, this is still one-step removed from the top step of Arnstein's ladder that defines empowerment as people taking control of decisions and their implementation [2].

Traditional public participation has centered on the gathering of citizens to discuss public issues in arenas such as town hall meetings or community advisory fora [3]. These methods of public participation required the public to meet in specific physical spaces. However, in the smart city era newer participation methods have emerged and these new methods could even transcend the need for physical gatherings. Five different participation approaches, including crowdsourcing, feedback forms, online voting serious games and immersive virtual environments, have been developed in various smart cities are discussed below. The first examples are related to collecting ideas from the public and are referred to as idea generation and the latter examples are related to the recommendation of the ideas and are referred to as idea evaluation [4].

Crowdsourcing is an online, distributed problem-solving model that is used in smart cities to collect information on a wide range of topics [5]. Typically, the aim is to collect a vast amount of ideas on how to improve the city, the ideas are generated by a large group of citizens, and the topics can extend to any component of the city. For example, the city of Zaragoza has developed online tools to receive feedback on the local public infrastructure. The city provides a list of the reports and faults that describe the current situation and they ask the local citizens for new complaints and suggestions. The information that is crowdsourced from the citizens can relate to any element of the public infrastructure and in this case the location of the comment is automatically attached to the feedback form through the geo-location capability of the device that the citizen uses [6]. In another example, geolocated crowdsourced data were used to find vacant spaces that were suitable for regeneration across a whole city [7].

Another smart city approach is to request feedback on a narrow range of topics that are of particular interest via feedback forms. In this case the problem is related to a specific component of the city and the feedback is once again given by the crowd. For example, the Spanish City of Castellón has developed an online feedback tool that allows users to give feedback regarding the city's bike sharing facilities. This was considered to be a suitable topic for feedback as the city had already created an app that reported the real-time availability of the bicycles at each station. The app allows the users to report issues relating to the sharing services and the state of the bicycles [6]. Similarly, Maptionnaire [8] can be used to collect feedback on a specific range of topics. It allows organizations to quickly create their own online questionnaires and the questionnaires can then easily be linked to the areas of interest via GIS-based maps. This enables the feedback to relate to specific areas of the city and to report on how it can be improved. The tool also has the capability to analyze the feedback and to store all of the collected citizen feedback in a common database.

Citizens can also engage with their city government on a specific set of topics via voting. This allows the public to read about a range of options and to recommend the option that they consider to be the best fit. An advantage of this approach is the speed with which citizens can participate as it does not require qualitative feedback. For example, the City of San Francisco has utilised the MindMixer platform and encourages participation through civic-based rewards. In one scenario it was used to select a new logo for the San Francisco Municipal Transportation Authority [9].

In some cases, serious games have been used to receive feedback from the public [10]. One urban planning example from Germany concerned the renovation of a number of university buildings. There were four possible options for the project which included the renovation of the existing buildings, demolition and new construction of the existing buildings, partial relocation to a new site or complete relocation to a new site. The goal of the game was to encourage the public to study the alternative options in detail rather than just speed-reading through the proposals. In order to be successful at the game the players needed to understand the advantages and disadvantages of the various options and to make tradeoffs when selecting their preferred option. The game communicated the tradeoffs and enabled the players to find the urban planning solution that was most acceptable to them. In a similar experiment, participatory games were used to generate content from tourists in order to collect "effective and reliable information resources for other tourists" [11].

A wide group of stakeholders can also participate as a group in immersive virtual environments. In another urban planning case, virtual reality proved to be useful for communicating complex spatial information to urban stakeholders. Using virtual reality in a workshop allowed stakeholders involved in all realms of urban infrastructure management to sit down together and discuss the interdependencies between the different areas of responsibility. For example, the stakeholders from the parks and recreation department interact were able to understand the impact that their plans would have on the stakeholder responsible for repairs of the water and sewage works. Virtual reality provided a "common language" for specific spatial phenomena to be discussed [12]. Virtual reality can also benefit public participation projects by enabling the citizens to explore the options in much more detail than in traditional geographical media with lower spatial resolutions such as static 3D images [13].

The use of online tools clearly reduces the barrier to participation. They reduce the effort needed to participate and they are important steps towards effortless or passive participation. The citizens are not required to report the feedback in person at an official location or at a community meeting and the expected result is that the public bodies will receive a greater amount of feedback. There are two reasons why the amount of feedback is increased. Firstly, they may increase the total number of citizens that participate in public dialogue and secondly, they may increase the frequency at which they participate over a certain period time.

It must be noted, however, that some scholars have highlighted the inherent disorderly nature of public participation and especially when digital tools are used [14]. For example, map based digital participation tools can empower some and marginalize others depending on technological and societal barriers [15]. The participation methods should be carefully designed so that the process can be understood by all of the citizens participating and even then, the results of any singular approach should be treated with healthy skepticism. It should be assumed that there is no one size fits all solution to public participation and experimentation with multiple types of participation simultaneously is recommended [14].

This article attempts to make three contributions. First, it adds to the smart city literature by further exploring the use of big data to aid public participation. Secondly, it adds to the literature on digital citizenship and surveillance by providing an empirical example of how passively generated data can be safely used to provide important insights to a municipal body. Finally, it adds to the environmental sustainability literature on the considerations that should be made when using personal data under the guise of reducing environmental impact. It aims to go beyond these newer smart city methods of participation. It examines the role of big data in public participation and asks; how can passively generated data be used to inform the design the public services of a city in a way that is inclusive, safe and accurate?

## 2. Data-Driven Public Participation

In addition to the traditional and the newer smart city participation methods described above, there has been a growing interest in using the data that is pervasively gathered in cities to understand the needs of citizens [16–18]. Some scholars have even argued that data is a common pool resource that can be collected to benefit society [19]. Hintz et al. [20] maintain that we are all digital citizens who increasingly interact with our social and political

environment through digital media and that digital tools and platforms have become essential in order to participate in society. We actively self-construct our digital citizenship by volunteering information about ourselves in the public sphere. However, regardless of our deliberate actions, we increasingly live and operate in a datafied environment in which everything we do leaves data traces. A large number of our activities online and increasingly offline, generate data. We generate geo-location data when we walk around with our mobile phones and our interactions on social media can give insight on who we are. Our digital citizenship is defined through our online actions and not by our official status of being a citizen of a nation state and the associated rights and responsibilities that are linked to that [20].

When we consider how our digital citizenship is created and maintained we notice that there is both an active and a passive component to this. Blogs or social media postings without privacy settings, such as a public twitter account, are simple examples of how people deliberately and actively self-construct their digital citizenship. Some scholars have referred to this kind of active contribution as being "volunteered" [16] and as being made by "conscious" user actions [17]. In the context of the smart city, active content is often created by members of the public to replace or enhance existing data sets such as Wikipedia, OpenStreetMap or fixmystreet.com [17]. Citizens also play an active role by using publicly available open data to create mobile and web applications that help other citizens such as journey planners or guidance for mentally disabled people [21,22]. This kind of active digital citizenship is also connected to citizen empowerment through examples such as online citizen journalism [23] and political activism via content on social media [24].

### 2.1. Active and Passive Participation

Active participation follows the logic of active digital citizenship in that it is deliberate and self-constructed. Whereas passive participation is an unintentional result of living in a datafied environment where our actions can be used to understand the needs of citizens. Passively generated data is effortless and often unintentional. It can be generated by completing simple tasks that seem minor relative to the complexity of our daily lives such as buying a bus ticket. However, insight may be gleaned by combining passive data from multiple sources. The passive component of digital citizenship views "citizens as sensors" [6,16,17]. This approach enables the opportunity for passive participation where data mining can be used to further elicit meaning from the data. The data will be extracted, aggregated, and analyzed by algorithms. This form of participation will be the product of harvested public opinion and could use sentiment analysis algorithms to determine the meaning and topic of the text of a tweet which in turn could be used within city level decision-making. At its full potential "city officials could effortlessly scrape public opinion from citizens' twitter feeds and interactions across the city" [18]. Accurate passive participation would be particularly helpful as city agencies see public participation as useful but difficult to obtain [25]. The online and data mining approach can also help to solve the problem that arises in public hearings, which often pit citizens against each other, and this in turn creates ambivalence towards public participation within public agencies [26].

However, there are clear challenges with using the data that is gathered in a city in order to understand the needs of the citizens. The data itself is fragmented, distributed, and often only implicitly inferable and it is scattered on various web sources and databases [20]. There are also difficulties in determining sentiment from informal sources such as social media and in deriving insights from wide ranging and incomplete data sets that on the surface are completely disconnected from the public services that they are influencing. This is not to say that they are unsuitable but rather to express concern of the maturity of the technologies that are expected to replace much more active and direct participation such as traditional citizen engagement.

There are also concerns with how the data will be used beyond the shaping of public services. It is clear that the collected data is highly sensitive and that many privacy issues are involved and for this reason it is crucial that users are informed about which data gets collected and how it is distributed. If our datafied environment, meaning all of the digital traces of the citizens in a city, is being used for public participation then its meaningful

enactment requires an informed and knowledgeable understanding of the technologies in place [20]. In response to this The city of Barcelona has published the "Barcelona Ciutat Digital: A Roadmap Toward Technological Sovereignty" which promote citizens' data ownership and technological sovereignty [27,28].

Zuboff [29] describes the problem with "surveillance capitalism" where citizen data is collected, stored, monitored, shared, and sold. The entities involved in this exchange are social media services, other online platforms, data brokers, intelligence agencies, and public administrations. The results of surveillance capitalism can extend beyond privacy infringements and towards citizen control. This citizen control is the opposite of control by citizens as described by Arnstein [1] but rather the control of citizens as the insights extracted from data can lead to behavioral prediction and eventually to behavioral modification [29].

### 2.2. Data-Driven Service Design and the Sharing Economy

Public bodies can learn from how commercial services are already using passively generated data to design their service offering. Data is needed to design the service offering in modern cities as "customers are increasingly demanding complex, sustainable, and integrated solutions rather than standardized and homogeneous products and services" [30]. The sharing economy, also called the access economy, has the potential to meet the need of these new customer expectations and passive participation data can play an intrinsic role in the development of the service offerings in that industry. The sharing economy includes businesses such as car sharing (Zipcar) and space sharing (WeWork) where a firm's assets are temporarily rented to consumers [30]. The sharing economy also delivers some of the sustainability targets of the smart city as the re-use can ease the pressure on natural resources. It is also thought that the shift from individual ownership to collaborative consumption can reduce hedonistic consumerism and provide a sustainability framework based on community sharing [31].

WeWork offer shared workspace and related services and it is an example of a sharing economy concept where the service offering is continuously reconfigured based on the digital traces of the customers. Buildings are typically designed as bespoke projects where the assortment of spaces is selected to meet the needs of the users. Once the interior walls are constructed, they are not expected to be demolished or moved for years to come. Similar to a family home, you cannot reconfigure the floorplan frivolously. If for example, a particular office building is designed to have a boardroom for twenty people and ten small meeting rooms, then that is the space offering that is available to users in the short to medium term. In contrast, WeWork have challenged this way of thinking and they are constantly adjusting and updating their space offering based on passively generated data. When the users want to choose a desk or a meeting room to work in, they have to reserve the space via an online reservation tool and this creates a data trail. If the building has a number of alternative open office areas with themes such as quiet library zone, noisy coffee house zone and jazz music zone then the users can choose the space that suits their working style on that particular day. The data gathered by sensors and the reservation system can then show which spaces they need more of and which spaces they need less of [32]. If there are ten phone boxes for making private phone calls and the data shows that the peak occupancy at any one time is 5 phone boxes then the chances are that 5 of them will be removed and replaced with a workspace that is more in demand.

A recent study on the design of workspaces has argued that achieving both lower costs and higher productivity requires taking a data-driven and holistic view of the workplace [33]. Apart from the commercial benefits of cost and space efficiency, there is also a case to be made that a continuous reconfiguration of the offering provides a co-created and inclusive set of services. This is because the remaining offering has been optimized based on the citizens needs and thus it is giving the citizens exactly what they want. The same author as above also argues that "the use of data and evidence to drive decision making must not be confused with centralization of control. It should be considered as the opposite, better data must drive decentralization of managerial control and must also be part of an extremely transparent way of working" [33].

## 2.3. Direct and Indirect Passive Participation

Lin [24] describes the creation of political content on social media as active public participation and that people who like the content are enacting passive public participation. Similarly, other scholars have studied how the datafied environment of a city will be increasingly used as a form of public participation [15–17]. These descriptions of passive participation raise certain questions regarding their suitability. Likes of posts on social media are often untrustworthy sources as these actions are linked to a range of factors other than people's opinion. For example, the motivation to like a particular post is strongly related to the number of likes it already has and whether your peers have already liked it [34,35]. In addition, passive reactions to social media are for many reasons unsuitable as a mode of citizen engagement. The options to like the post or to ignore it pales in comparison to the minimum engagement level of agree, disagree, cannot say. Furthermore, social media likes cannot be taken seriously when they are surrounded by updates and photographs of children's birthdays, sporting achievements, holidays and other social occasions. The concerns around privacy and accuracy from the datafied environment are compelling cases to reconsider data mining for passive participation at today's technological capabilities. In this article, these examples are classified as indirect passive participation. The data could be used to interpret the needs of citizens but there is not a clear and trustworthy link between the data creation and the use of the data.

Direct passive participation should have a clear link between the data creation and the use of the data, the data collection process should be transparent and it should be collected without privacy concerns for the citizens. For example, if I buy a bus ticket at 8 a.m. on my way to work then I am in a way voting for that bus. My usage of that bus says that I value its existence and if I use that particular bus stop then it is more relevant to my journey than all the others in the city at that time. This is similar to the data-driven design of workspaces by Wework described above [33]. Table 1 below displays the range of public participation methods in the context of the smart city and positions direct passive participation, shown here as data-driven, relative to the other examples. Whereas, Figure 1 below shows the active and passive public participation methods relative to the time invested to participate and the directness of the data. The directness of the data is a measure of the link between the creation and the use of the data and serves as a proxy for the accuracy and trustworthiness. The time invested to participate is important with the assumption that lowering the barrier to participate will increase participation across all citizen groups thus increasing inclusiveness.

**Table 1.** The dimensions of public participation methods in the context of the smart city.

| Mode | Data | Realm | Initiator | Method | Short Description | Reference |
|------|------|-------|-----------|--------|-------------------|-----------|
| Active | Direct | Physical | City | Traditional | Town hall meetings or community advisory fora | 3 |
| Active | Direct | Digital | City | Crowdsourcing | Collect information on a wide range of topics | 5, 6 |
| Active | Direct | Digital | City | Feedback forms | Request feedback on a narrow range of topics | 6, 8 |
| Active | Direct | Digital | City | Online voting | Recommendations on a specific set of topics | 9 |
| Active | Direct | Digital | City | Serious games | Study the alternative options in detail & provide feedback | 10 |
| Active | Direct | Digital | Citizen | Information sharing platforms | Tools for other citizens to populate with information | 15 |
| Active | Direct | Digital | Citizen | City apps | Citizen created create mobile and web applications | 18, 19 |
| Active | Direct | Digital | Citizen | Citizen journalism | Citizens collecting evidence instead of journalists | 20 |
| Active | Direct | Digital | Citizen | Online activism | Creating political posts on social media | 21 |
| Passive | Indirect | Digital | Citizen | Social media likes | Liking a political post on social media | 21 |
| Passive | Indirect | Digital | City | Data mining | Data mining of indirect data sources | 14–16 |
| Passive | Direct | Digital | City | Data-driven | Using direct data sources | 28 |

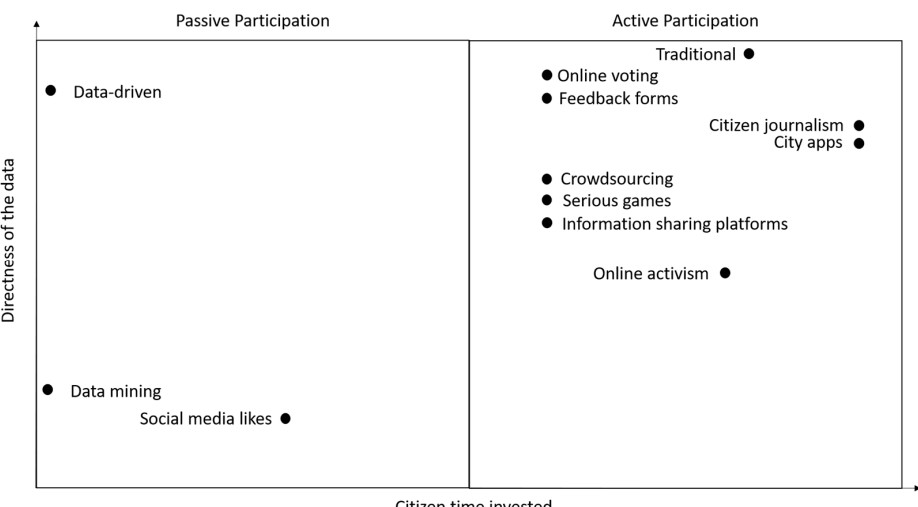

**Figure 1.** Active and passive public participation methods illustratively shown relative to the time invested to participate and the directness of the data.

## 3. Direct Passive Participation from a City Bike Sharing Scheme

### 3.1. Research Process

The Helsinki city bike share scheme has been selected as a case study to further explore the concept of direct passive participation. The scheme was set up in 2016 by the Helsinki Regional Transport Authority and in its first year it comprised of 50 stations with a total of 500 bikes. By 2019 it had grown to 308 stations and 3080 bikes. This case is closely related to Sustainable Development Goal 11 as it concerns a basic public service and one that is sustainable and encourages citizens to lead healthy lifestyles. The available data is from 2017 to 2019 and the aim is to uncover if passive user generated data from within the service can be used to reconfigure the service as part of a process of continuous development in order to be resource efficient and to meet the needs of the citizens. The data can also potentially provide the inclusiveness that is part of the requirements of SDG 11. The city bike scheme is operated by the Helsinki Regional Transport Authority who state that they "collect customers' personal data in order to provide smooth and effective transport services" [36]. The joint local authority is made up of the municipalities are Helsinki, Espoo, Vantaa, Kauniainen, Kerava, Kirkkonummi and Sipoo. It is a public agency and therefore it complies strictly with EU General Data Protection Regulations (GDPR). Customer data is only used for and only uses the data collection for internal purposes and is not usually shared with third parties. Personal data are only disclosed to third parties when obliged by law or if explicit consent has been given to do so. A case study research design has been selected as the core data set involves processes, activities and events. It also includes complex events and behavior which occur in a real-life setting which also makes it relevant to the case study approach [37,38].

The research is based on a number of resources. The primary data source is provided by the Helsinki Regional Transport Authority as open data and consists of the status of each station at 5 min intervals and details on each individual trip taken with a shared bike. More details on these datasets are provided below. Once the primary data source had been analyzed, a 60 min online workshop was held with three key stakeholders to discuss the initial findings. These stakeholders were made up of the project manager for the city bike scheme for Helsinki City Transport, another employee for Helsinki City Transport who is responsible for the service development of the city bike scheme and a representative of Helsinki Regional Transport Authority who is responsible for GIS data and open data.

The data regarding the status of each station is recorded every 5 min and comprises of the station identification code, station name, station location via coordinates, number of bikes available, number of spaces available and the timestamp of the data. An example is shown below in Table 2 for station 9.

**Table 2.** Example of the station status data.

| Station ID | Station Name | Latitude | Longitude | Bikes Available | Spaces Available | Timestamp |
|:---:|:---:|:---:|:---:|:---:|:---:|:---:|
| 9 | Erottajan Aukio | 60.167 | 24.944 | 27 | 3 | 15/08/2017 09.25:02 |
| 9 | Erottajan Aukio | 60.167 | 24.944 | 30 | 0 | 15/08/2017 09.30:02 |
| 9 | Erottajan Aukio | 60.167 | 24.944 | 30 | 0 | 15/08/2017 09.35:02 |

The data regarding each trip made by the users comprised of departure time, return time, departure station identification code and return station identification code. An example is shown below in Table 3 and please note that no details of the user who made the trip were made available.

**Table 3.** Example of the data generated for each trip.

| Departure Time | Return Time | Departure Station ID | Return Station ID |
|:---:|:---:|:---:|:---:|
| 30-05-2017 23:59:00 | 31-05-2017 00:06:00 | 160 | 220 |
| 30-05-2017 23:54:00 | 30-05-2017 23:58:00 | 120 | 129 |
| 30-05-2017 23:52:00 | 30-05-2017 23:57:00 | 800 | 300 |

*3.2. Research Methods*

The target of the research was to calculate the optimum size of the city bike stations using direct passive participation data and to compare this to the method of station sizing that was already in use. This way the accuracy of the data-driven approach could be compared with the traditional method which is described below. Finding the optimum size of the stations was interesting to the service designers as the stations are removed each winter and reinstalled each spring. This meant that the station size could be reconfigured each year without a large amount of additional work and thus they could be proactive with data better meeting the needs of the citizens.

Each city bike trip begins by unlocking a bike at a station and the bike must be securely returned to a station with the network at the end of the trip. The size of the stations is measured in slots and each slots fits one bike. If all the slots are taken, a bike can be returned to the station by locking it to another bike which is already secured. The maximum number of slots in any one station is 60 and the minimum number is 12. During the workshop it was revealed that a station that is expected to have an average demand will be allocated 20 slots.

The existing methods of sizing the station included rules of thumb and a review of the data at the end of each year. The rules of thumb included studying the best practice from other countries and details of the area surrounding each individual station such as population density, workplace density, the existing traffic network and cursory knowledge of the demand of public transport in that area. The structure of the existing traffic network is relevant as the bike stations should be located on busy roads to maximize visibility and access. At the end of the 2019 season a number of days were used to analyze that year's data with the aim of improving the service and the sizes of 60 of the 3080 stations were changed. The workshop revealed that they had not been exhaustive in their use of data. The public bodies trusted the original rules of thumb and the purpose of the review was to look for any clear mismatches between the demand, the characteristics of the area and the size of the stations. It was clear that a time-consuming exhaustive study using data to define the needs of each city area had not been undertaken. This detailed approach was undertaken as part of the research described in this article.

The approach of the research described in this article was to compare the demand of each station to that stations size and to do this for all stations to find the correct balance between demand and size. Therefore, if one station had x number of departures in a given time period then it would be expected to have half the number of slots of a station with twice as many departures. The data from May to August 2019 was used as new stations were added between the 2018 and 2019 season and this was the latest year that included all of the stations. May to August was selected as 70.5% of all trips in the 9 month city bike

season occurred during these 4 months. The average number of departures per slot was calculated for the whole network during this time period. Any station with a departures per slot ratio below this was considered to be oversized and any station with a departures per slot ratio above this was considered to be undersized.

The demand calculation was then used to calculate the optimum size of the stations. The total number of slots in the network and the locations of the stations were fixed and the optimum number of slots per station was calculated by dividing the total number of departures from that station by the average number of departures per slot in the whole network. Departures from a station were used instead of using both departures from and returns to that station in order to simplify the calculation and as the location of the departure station is the primary driver for using the bikes. In other words, people spontaneously use the bikes when they are easily accessible and the proximity at the start of the journey is more important than the proximity of the drop off station to the destination.

## 4. Results

The removal and return of a city bike by a citizen leaves a digital footprint and this data can be used to indicate the components of the city bike systems that are meeting the needs of that citizen at that time. The station status data reveals how many slots are in that station during the different time period. The number of slots does not change often but is important to track the difference from year to year. The total number of citizen trips studied was 9.2 million rows of data and in the most recent year of 2019 it was 3.7 million. The total number of records of station status was 48.8 million rows of data and in the most recent year of 2019 it was 26 million. The main purpose of the station status data is to monitor how many slots are in each station as the station sizes changed frequently in the early years of the scheme.

The average number of departures per slot was calculated for the whole network during this time period. Any station with a departures per slot ratio below this was considered to be oversized and any station with a departures per slot ratio above this was considered to be undersized. This approach clearly compares the citizen demand to the service offering and can be used to reconfigure the size of the city bike stations.

The proposed adjustment in the size of the stations can be seen in Figure 2 below. The data used in the calculation included all trips from Monday to Friday during the hours of 08.00–11.00 and 15.00–18.00 for May–August 2019. The blue and red circles indicate the locations of the stations and the size of the circles indicates the magnitude of change that is proposed. The magnitude of change is a measure of the number of slots that should be added or removed. The stations in red are recommended to grow in size and the stations in blue are recommended to reduce in size. These results were presented to the stakeholders during the workshop. At this point it was revealed that the most important times of the week for the operation of the city bikes are Monday to Friday from 08.00–11.00 and 15.00–18.00. This is in order to alleviate the congestion caused by the morning and evening rush hour. The calculation was then redone using only the trips made during these hours.

The calculation model suggests a new size for every station in the network. Figure 3 below displays the details of one particular bike station called Kalasataman Metroasema. Each circle in the visualization displays 6 parameters when selected. These parameters include the sign of the change (1 indicates that the station should grow, −1 indicates that the station should reduce in size) and the stations latitude and longitude. Other parameters are the change in the number of slots and in this case the model suggests that the station should grow by 46.81 slots. The name of the station and the existing number of slots are also displayed. This visualization can be shown for any of the 3080 stations.

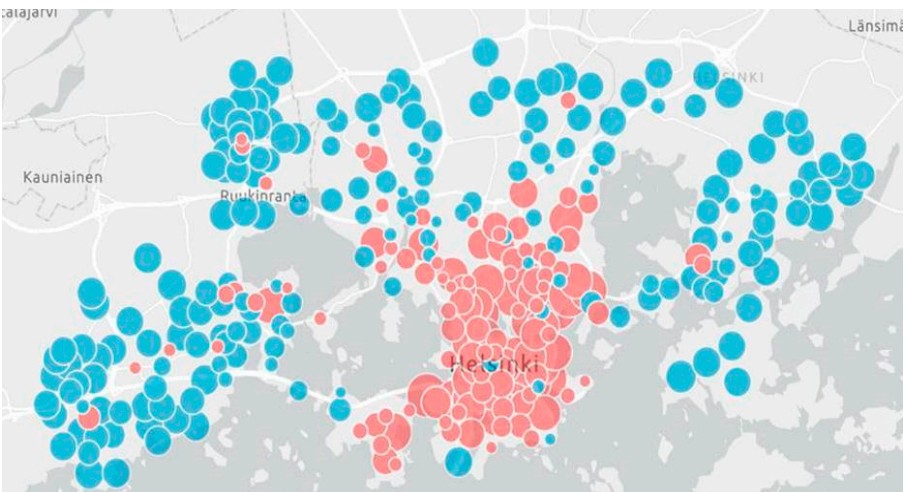

**Figure 2.** The proposed adjustment in the size of the stations using data from direct passive participation.

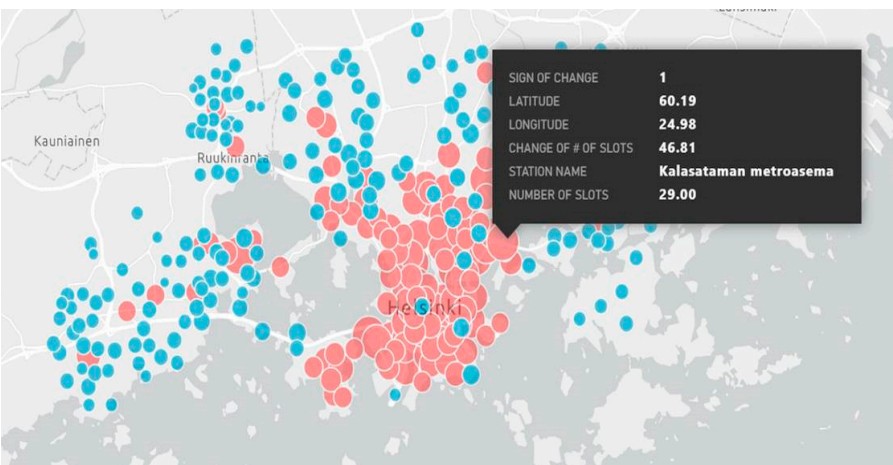

**Figure 3.** The information that is connected to each circle in the calculation model.

## 5. Discussion

Rules of thumb such as population density, workplace density, the structure of the existing traffic network and some demand data from the local modes of public transport are sensible choices in the first years of the service in the absence of direct demand data. They can help to design the service in the first year to test demand and to start collecting dynamic data on which stations are popular and why. The concept of starting small and using data and customer feedback to develop the service is common approach in software development and is widely used by startups in all industries [39]. Using dynamic data is similar to the approach used in Korea [40] when designing local city services which they called local network development. However, after the first year the original rules of thumb should be reconsidered. Population density and workplace density may be good measures of how many people are likely to be in that area, they are not directly related to the demand for modes of public transport such as city bikes. For example, one neighborhood may have a higher proportion of car ownership than another largely due to socio-economic factors. An area with high workplace density near the center of a city may have a high percentage of commuters using public transport due to congestion and high parking fees whereas the opposite may be true for an area of with a similarly high workplace density on the outskirts of a city. Thus, the historic demand data collected from the city bike stations can be considered a key indicator in the design of the service.

This case is an example of how passively generated data can be used to design the public services of a city in a way that is inclusive, safe and an accurate representation of the citizen's needs. It is an example of the method shown as data driven in Figure 1 which provides the optimum balance between accuracy for the city and effort for the citizen. It is different from the many of the other dimensions of smart city public participation methods shown in Figure 1 as it is passively and not directly generated. If is different from the other passive examples referred to as data mining and social media likes as the data is related to the problem at hand.

The passiveness is ensured by that fact that each participative data point is created without any extra effort by the users of the system. This also delivers the inclusiveness as each user of the system has an opportunity to participate in the design of the future version of the service. The safety is provided by the fact that the data is also fully anonymous. When a citizen borrows a city bike the system records when and where that particular bike has been borrowed from and the same is done when the bike is returned. The demographics of the citizen borrowing the bike are considered irrelevant in order to understand the optimum size of the stations. The accuracy is provided by the directness of the data and there is an undeniable link between the creation and the use of the data. The 308 city bike stations are distributed all over the city. However, the size of each individual station is the most important element of the service provision by the city to the citizens. It defines which areas in the city will have larger or smaller stations and thus which areas will get more or less bikes than average. The city focuses on the data generated during Monday to Friday from 08.00–11.00 and 15.00–18.00 as one of the main purposes of the city bike system is to relieve pressure on the transport infrastructure during the morning and evening rush hours. Hence, the demand during these times is the primary source.

## 6. Conclusions

Many scholars have made compelling cases as to why the datafied environment of a city will be increasingly used as a form of public participation [15–17]. However, there are valid concerns regarding the ability of existing technology to interpret the data which is often fragmented and implicitly inferable [16]. In addition, in the quest for Arnstein's [1] control by citizens we may be opening the door to control of citizens as the insights extracted from the datafied environment can lead to behavioral prediction and eventually to behavioral modification [29].

This article has built upon previous literature and has highlighted two key characteristics of using data from digital citizenship. The first is the potential of passively generated data and the second is how relevant the data is relative to the problem it is being used to solve. Data can be actively or passively generated. Actively generated data is deliberate and self-constructed whereas passively generated data is an unintentional biproduct of living in a datafied environment. Passively generated data is created effortlessly and thus has the potential to provide a huge amount of data that can inform the design the public services of a city. It widens the number of people that will engage in public participation.

The relevance of the data to the problem at hand should also be considered. Direct data should have a clear link between the data creation and the use of the data. If the question at hand concerns whether or not to build a park in neighborhood A, then an example of a direct data source would be the opinions of the local stakeholders on the proposed development. Indirect data sources are not directly related to the problem at hand and require extrapolation in order to aid the analysis. It could for example, be related to the demographics of neighborhood A or the number of residences in the neighborhood that have their own back yard. It is possible that these indirect sources can be helpful but they are not the ideal source. Indirect passive participation is an important area of development and as it is being developed, direct passive participation can already be used as one of many tools to add inclusiveness to a city's public services. Direct passive participation should have a clear link between the data creation and the use of the data, the data collection process should be transparent and it should be collected without privacy concerns for the citizens. This makes the data more trustworthy in terms of accuracy and safe in terms of the risk of behavioral modification.

Positive steps are being taken to promote the safe use of citizen data. One example is MyData Global, which aims to empower individuals by improving their right to self-determination regarding their personal data and the nonprofit organization has members in over 40 countries [41]. Their principles have been developed to enable citizens to transparently see how their data is used, to control how their data are used and to claim their share of the benefits in sharing of their data. If MyData succeeds, then there is still a need to understand how indirect data can be used to serve the needs of citizens. The indirect data will extracted, aggregated, and analyzed by algorithms and it is expected that these processes will improve over time. Thus, the method shown as data mining in Figure 1 above will begin to move up the *Y*-axis and become more accurate and trustworthy. If the goal of city governments is to inclusively provide housing, transportation and other basic services then they need direct, regular and democratic participation of citizens in order to do so. At present, the dream of scraping accurate public opinion from citizens twitter feeds and their daily interactions across the city is still very far from reality.

The case study described in this article relies on data from customers to help customers [34]. It shows a method in which the existing design and suitability of public services can be reassessed and reconfigured over its lifecycle using dynamic data that is passively user generated. A valid concern in the use of direct passive participation approach in this article is raised by Chandler [42] who studies how big data can potentially empower marginal and vulnerable people. People who have not datafied their daily life and thus are not digital citizens should be considered when aiming to create participation methods that are widely inclusive. Similarly, people who do not own smart phones and who cannot vote for city services with their purchases should not be left out of the process. We need to create room in our analysis for citizens who do not take the bus or use the city bikes because they are forced for financial reasons to walk or hitch a ride. It is also important to question machine readable data points as being the most important component of participation. Other methods should be considered in order to engage a wide range of citizens including those not adept at making complex arguments. For example, one study experimented with participation via decision theatre, activist art and online games [43].

**Funding:** This research received no external funding.

**Acknowledgments:** Thank you to Anna Korolyuk who was responsible for extracting and visualizing the data and helping with the analysis.

**Conflicts of Interest:** The authors declare no conflict of interest.

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
