# Peer review of "Direct Passive Participation: Aiming for Accuracy and Citizen Safety in the Era of Big Data and the Smart City"

_smartcities, doi:10.3390/smartcities4010020_

Round 1

Reviewer 1 Report

This manuscript could be an interesting contribution to the broad field of direct passive participation. The authors show that they are familiar with the current debates and that they can contribute to several up-to-date topics in the important field of (big) data acquisition and use in smart cities. Although the authors declare three aims at the end of their long introduction, the focus of this manuscript appears to be on an empirical investigation. I think this manuscript would enrich the ongoing debates and I would like to encourage the authors to consider the following two points which might contribute to publishing an improved version of the current manuscript:

  • Could you please be more precisely when deriving recommendations from your case study? Toward the end of your paper you give several “We must” statements that built-upon your study. Could you link them a bit more to the current debates (literature references) so that readers could see the connections within the ongoing debates?

  • When introducing the idea of serious games in planning (l. 102), the authors should mention recent methodological developments. The opportunities of the modern field of immersive virtual environments (Virtual Reality) has been under study in the last years. Authors should refer to this form of modern and interactive form of spatial data presentation. These papers could serve as references:

Edler, D., Keil, J., Wiedenlübbert, T., Sossna, M., Kühne, O., Dickmann, F. (2019). Immersive VR Experience of Redeveloped Post-Industrial Sites: The Example of "Zeche Holland" in Bochum-Wattenscheid. In: KN – Journal of Cartography and Geographic Information, 69 (4), pp. 267-284. https://doi.org/10.1007/s42489-019-00030-2

Ma, Y., Wright, J., Gopal, S., Phillips, N. (2020). Seeing the invisible: From imagined to virtual urban landscapes. In: Cities, 98, 102559. https://doi.org/10.1016/j.cities.2019.102559

Author Response

Dear reviewer thank you for your helpful comments.

  1. The results section has been rewritten and the results and discussion into separate sections. The empirical data is now more clearly and more comprehensively presented.
  2. The suggested references have been added and they fit very nicely. Thank you for highlighting them. I have added 6 new references in total which are numbers 12,13,19,27,28 and 36.
  3. The recommendations of the study have been made clearer in the discussion starting in line 501 and this now links better to the rewritten research question. The main results are the discussion on (a) the potential of passively generated data and (b) the concept of how relevant the data is relative to the problem it is being used to solve.
  4. The results serve to show that when data is generated passively and therefore effortlessly it the potential to provide a huge amount of data that can inform the design the public services of a city. It widens the number of people that will engage in public participation.
  5. The results serve to show a case where the passively generated data is relevant to the problem at hand. Direct data should have a clear link between the data creation and the use of the data.
  6. The results are a clear example of passive and direct data which this article recommends in order to be inclusive, safe and accurate.
  7. The "we must" statements have been toned down and changed to the form of "these should be considered"

Reviewer 2 Report

This paper presents a good review for the passive participation in smart cities. A bike sharing case study is used to demonstrate the effect of passive participation. Some comments are listed as below:

(1) How to demonstrate the performance of the passive participation? The quantitative results should be provide to illustrate the performance of passive participation in smart cities application. Furthermore, the comparison with other methods is expected.

(2) How to deploy the passive participation in smart cities applications?

(3) Would you please provide more passive participation method expect social media likes?

Author Response

Dear reviewer thank you for your helpful comments.

  1. The results section has been rewritten and the results and discussion into separate sections. The empirical data is now more clearly and more comprehensively presented.
  2. The results serve to show that when data is generated passively and therefore effortlessly it the potential to provide a huge amount of data that can inform the design the public services of a city. It widens the number of people that will engage in public participation.
  3. The results serve to show a case where the passively generated data is relevant to the problem at hand. Direct data should have a clear link between the data creation and the use of the data.
  4. The results are a clear example of passive and direct data which this article recommends in order to be inclusive, safe and accurate.
  5. The article has not described how the data can be used in smart city application as the focus has been on the kind of data characteristics that can deliver inclusive, safe and accurate participation.
  6. The example of social media likes has been used to demonstrate an example of public participation that is used but is not recommended by this study. The reference is from a scientific article. If it is not suitable for this article then it can be completely removed.

Reviewer 3 Report

Interesting article that demonstrates the importance of data for urban planning.
I wish I had seen the research question.
I also missed reading about Elinor Ostrom which demonstrates that data is a common good when used for the good of society.
As for data security, the author mentions, but I was curious to know what security guarantee the Finns receive when they join the service.

Author Response

Dear reviewer thank you for your helpful comments.

  1. The research question has been rewritten and now links better to the recommendations of the study in the discussion. It can be seen in line 151.
  2. The recommendation have also been rewritten starting in line 501. The main results are the discussion on (a) the potential of passively generated data and (b) the concept of how relevant the data is relative to the problem it is being used to solve.
  1. The results serve to show that when data is generated passively and therefore effortlessly it the potential to provide a huge amount of data that can inform the design the public services of a city. It widens the number of people that will engage in public participation.
  2. The results serve to show a case where the passively generated data is relevant to the problem at hand. Direct data should have a clear link between the data creation and the use of the data.
  1. The results are a clear example of passive and direct data which this article recommends in order to be inclusive, safe and accurate.
  2. The reference to Elinor Ostrom has been added and it fits very nicely. It also inspired two new references related to the work of Francesca Bria in Barcelona.
  3. Approximately 8 lines have been added regarding the privacy policy of the local authority running the city bike scheme. A link to their privacy policy has also been added.